# Intercropping with Potato-Onion Enhanced the Soil Microbial Diversity of Tomato

**DOI:** 10.3390/microorganisms8060834

**Published:** 2020-06-02

**Authors:** Naihui Li, Danmei Gao, Xingang Zhou, Shaocan Chen, Chunxia Li, Fengzhi Wu

**Affiliations:** 1Department of Horticulture, Northeast Agricultural University, Harbin 150030, China; linaihui1992@aliyun.com (N.L.); dmgao2019@neau.edu.cn (D.G.); xgzhouneau@gmail.com (X.Z.); yuhongjie2016@aliyun.com (S.C.); lcx198238@163.com (C.L.); 2Key Laboratory of Cold Area Vegetable Biology, Northeast Agricultural University, Harbin 150030, China

**Keywords:** intercropping, tomato, potato-onion, soil microbial community, Illumina MiSeq sequencing

## Abstract

Intercropping can achieve sustainable agricultural development by increasing plant diversity. In this study, we investigated the effects of tomato monoculture and tomato/potato-onion intercropping systems on tomato seedling growth and changes of soil microbial communities in greenhouse conditions. Results showed that the intercropping with potato-onion increased tomato seedling biomass. Compared with monoculture system, the alpha diversity of soil bacterial and fungal communities, beta diversity and abundance of bacterial community were increased in the intercropping system. Nevertheless, the beta-diversity and abundance of fungal community had no difference between the intercropping and monoculture systems. The relative abundances of some taxa (i.e., Acidobacteria-*Subgroup-6*, *Arthrobacter*, *Bacillus*, *Pseudomonas*) and several OTUs with the potential to promote plant growth were increased, while the relative abundances of some potential plant pathogens (i.e., *Cladosporium*) were decreased in the intercropping system. Redundancy analysis indicated that bacterial community structure was significantly influenced by soil organic carbon and pH, the fungal community structure was related to changes in soil organic carbon and available phosphorus. Overall, our results suggested that the tomato/potato-onion intercropping system altered soil microbial communities and improved the soil environment, which may be the main factor in promoting tomato growth.

## 1. Introduction

In greenhouse cultivation, the increase of the vast productivity in modern agriculture often comes at the price of sustainable agricultural development [1]. This is because modern agriculture can reduce biodiversity [2], which is the key factor influencing the functioning and stability of agro-ecosystems. Agricultural studies have found that intercropping can enhance biodiversity by increasing spatial and temporal plant diversity [3]. Moreover, intercropping systems have obvious advantages in ecosystem services, more so than those of modern agriculture, characterized by monoculture, such as efficient resource utilization and yield advantage [4]. Therefore, understanding the effect of intercropping management on plant growth and soil ecosystem can promote the development of new strategies for sustainable agriculture.

As an important part of the agro-ecosystem, the soil microbial community can be driven by many biotic and abiotic factors, including soil chemical properties, plant functional diversities and management practices, i.e., cropping system, irrigation and fertilization [5,6]. Considerable evidence has been showing that diversified cropping systems have higher soil microbial abundance and diversity and alter dominant soil microbial taxa and communities [7]. Furthermore, several reports have shown that diversity and composition of soil microbial community are critical to the maintenance of soil health and productivity [8,9]. In particular, some taxa could directly or indirectly affect soil productivity by producing plant growth hormones, solubilizing soil P and enhancing nitrogen fixation [10], such as *Bacillus* [11], *Pseudomonas* [12] and arbuscular mycorrhizal fungi (AMF) [13].

Tomato (*Solanum lycopersicum* L.) is an economically important crop in many countries and is commonly planted in monoculture in greenhouse production systems. Potato-onion (*Allium cepa* L. var. *aggregatum* G. Don), a variant of onion, has been regarded as a promising intercropping plant by some researchers, especially the variety “Suihua”, which has shown stronger positive effect in diversified cropping systems [14,15]. In our previous studies, we found that the tomato/potato-onion intercropping system was a diversified cropping system with multiple agro-ecological functions, such as alleviating the incidence and severity of tomato verticillium wilt [16], increased the community structure and function of phosphobacteria in tomato rhizosphere, and promoted the P uptake of tomato [17]. However, in this intercropping system, some other ecological functions are unknown.

In this study, the shoot and root biomasses of tomato seeding, soil physicochemical properties and microbial communities in tomato monoculture and tomato/potato-onion intercropping, were investigated. The soil microbial community abundance and composition were analyzed by quantitative PCR (qPCR) and Illumina MiSeq sequencing, respectively. We aimed to evaluate the potential influences of the tomato/potato-onion intercropping system on the tomato growth and soil microbial communities. We hypothesized that (1) the intercropping system would improve tomato growth, (2) the intercropping system had higher soil microbial community diversity and abundances than those in the monoculture system and (3) the intercropping system would exhibit higher relative abundances of bacterial and fungal taxa with the potential to promote plant growth and/or inhibit soil-borne pathogens.

## 2. Materials and Methods

### 2.1. Greenhouse Experiment

This study was carried out in the greenhouse located at the Xiangyang experiment farm of Northeast Agricultural University in Harbin, China (45°76′N, 126°92′E) from April to July in 2016. In the spring, 38,000 kg ha^−1^ decomposed manure (0.5% N, 0.5% P and 0.4% K) and 150 kg ha^−1^ diammonium hydrogen phosphate were applied as basal fertilizer. The field soil was black soil (Mollisol) with sandy loam texture. Before this study, watermelon had been cultivated in the greenhouse for one year. The initial soil sample was collected before the fertilization and the soil chemical properties were as follows: Organic matter, 2.68%; inorganic nitrogen (NH_4_^+^-N and NO_3_^−^-N), 121.67 mg kg^−1^; available potassium (AK), 194.67 g kg^−1^; available phosphorus (AP), 54.07 mg kg^−1^; EC, 0.25 mS cm^−1^; and pH, 6.15.

Tomato (*Solanum lycopersicum* L.) cultivar “Dongnong 708” and potato-onion (*Allium cepa* L. var. *aggregatum* G. Don) cultivar “Suihua” were used in this study. Two treatments were included: Tomato monoculture system and tomato/potato-onion intercropping system. A randomized block design was followed with three replicated plots (each 4.8 m long and 3.0 m wide). Tomato seedlings with five leaves were transplanted to each plot on 23 April 2016. There were five rows in each plot with 68 tomato plants. The tomato plants in the middle three rows were the experimental rows, and the outer two rows served as protective rows. In the intercropping system, three potato-onion bulbs were simultaneously planted on the side of every tomato. Drip irrigation was installed, and weeds were manually removed once every week after transplanting.

### 2.2. Plant and Soil Sampling

The plants and soil samples of the monoculture system and intercropping system were collected 45 days after transplantation. Twenty-seven tomato plants were randomly selected in the experimental rows in each plot, and carefully harvested. To collect the bulk soil samples, the tomato roots were gently shaken. Twenty-seven bulk soil samples (10 cm diameter, 20 cm depth) were collected from each plot of the monoculture and intercropping systems. The soil samples were collected and gently mixed to form a composite sample, sieved through a 2 mm mesh to remove the stones and roots and thoroughly homogenized. Thus, there were three composite soil samples for each cropping system. One part of each fresh soil sample was air-dried (<30 °C) for soil physicochemical properties analysis, and the other part was stored at −70 °C for DNA extraction.

The tomato plants were washed, and moisture from the root surface was absorbed by absorbent paper before measuring dry weight. Then, the clean tomato plants were dried to a constant weight in an oven at 75 °C. The dry weights of the tomato plants were measured on a scale (±0.001 g).

### 2.3. Soil Physicochemical Properties Analysis

Soil physicochemical properties were analyzed based on the methods described in Bao (2005) [18]. In brief, soil pH and electrical conductivity (EC) were determined with a glass electrode after suspending 10 g soil in a 25 mL deionized water suspension (1:2.5, *w*/*v*). Soil available phosphorus (AP) and inorganic nitrogen (NH_4_^+^-N and NO_3_^−^-N) were extracted with 0.5 M NaHCO_3_ and 2 M KCl, respectively, and then analyzed with a continuous flow analyzer (San++, SKALAR, Netherlands). Soil available potassium (AK) was extracted with 1.0 M NH_4_Ac and quantified by inductively coupled plasma-atomic emission spectrometry (ICPS-7500, Shimadzu, Japan). Soil organic carbon (SOC) was extracted using K_2_Cr_2_O_7_ and H_2_SO_4_ to digest 0.5 g soil, and the residual K_2_Cr_2_O_7_ was titrated with FeSO_4_•7H_2_O. Soil moisture contents were determined by drying to constant weight at 105 °C.

### 2.4. Soil DNA Extraction

Total soil DNA was extracted from 0.25 g frozen soil (wet weight) using the PowerSoil DNA Isolation Kit (MO BIO Laboratories, Carlsbad, CA, USA) according to the manufacturer’s instructions. Each composite soil DNA sample was extracted three times, and the three samples were mixed to make composite DNA samples. The composite DNA samples were stored at −20 °C for further analysis.

### 2.5. Quantitative PCR (qPCR)

Total bacterial and fungal abundances of the soil samples were estimated in triplicate by qPCR assays in an IQ5 real-time PCR system (Bio-Rad Lab, Hercules, CA, USA). For the bacterial communities, the primer set 338F/518R [19] was used to amplify the targeted bacterial 16S rRNA genes. Briefly, each 20 μL PCR reaction contained 9 μL of 2 × Real SYBR Mixture, 0.4 μL of each primer (10 μM), 2.5 μL of template DNA and 8.1 μL of ddH_2_O. The qPCR reaction conditions were as follows: 5 min at 95 °C for initial denaturation, 27 amplification cycles of 50 s at 95 °C for denaturation and 45 s at 62 °C for annealing. For the fungal communities, the primer set ITS1F/ITS4 [20] was used to amplify the targeted fungal ITS1 region of the genes. Briefly, each 20 μL PCR volume contained 9 μL of 2 × Real SYBR Mixture, 0.5 μL of each primer (10 μM), 2 μL of template DNA and 8.5 μL of ddH_2_O. The qPCR reaction conditions were as follows: 5 min at 94 °C for initial denaturation, 30 amplification cycles of 1 min at 94 °C for denaturation and 45 s at 58 °C for annealing. Sterilized water instead of soil DNA was used as negative control, and all the composite soil samples were analyzed in triplicate. The standard curves of bacterial and fungal were created with 10-fold dilution series of plasmids containing the target genes from soil samples, separately. The bacterial and fungal standard curve amplification efficiencies were 104.59% and 99.39%, respectively, and the R^2^ values of the standard curves were 0.998 and 0.999, respectively. The initial copy numbers of bacterial and fungal target genes were obtained with threshold cycle (Ct) values and standard curves, respectively.

### 2.6. Illumina MiSeq Sequencing and Data Analysis

The bacterial and fungal communities were analyzed using the primers 515F/907R [21] and ITS1F/ITS2 [22], which were used to amplify the V4-V5 region of the bacterial 16S rRNA genes and the ITS1 region of the genes, respectively. Both primers were modified with a unique 6 bp barcode at the 5′ end, which was used to identify each sample. The DNA samples were amplified using an ABI GeneAmp^®^ 9700 PCR System (ABI, Waltham, MA, USA) in 25 μL reactions containing 4 μL of 5 × FastPfu Buffer, 2 μL of 2.5 mM dNTPs, 0.8 μL of forward and reverse primers (5 μM), 0.4 μL of FastPfu Polymerase (Transgen Biotech, Beijing, China), 1.0 μL of template DNA (10 ng) and 16 µL of ddH_2_O. The PCR conditions were as follows: 3 min at 95 °C for initial denaturation, 27 amplification cycles of 30 s at 95 °C for denaturation, 30 s at 55 °C for annealing, 45 s at 72 °C for elongation, 10 min at 72 °C for a final extension for the 16S V4-V5 rRNA genes and 3 min at 94 °C for initial denaturation, followed by 35 amplification cycles of 30 s at 94 °C, 30 s at 55 °C and 45 s at 72 °C, and a final extension 10 min at 72 °C for ITS genes. Each composite sample was amplified in triplicate. The products of the PCRs were pooled and checked in a 2% agarose gel under UV light and then pooled and purified using the AxyPrepDNA Gel DNA purification kit (AXYGEN). According to the preliminary agarose gel results, the PCR products were quantified using the QuantiFluor™–ST Blue Fluorescence Quantitation System (Promega) and mixed according to the sequencing requirements for each sample. Then, the mixture was paired-end sequenced (2 × 300 bp) on an Illumina MiSeq platform at Majorbio Bio-Pharm Technology Co., Ltd., Shanghai, China.

The raw sequence files were quality filtered and processed using FLASH [23] as previously described [9]. The operational taxonomic units (OTUs) were clustered at 97% sequence similarity using UPARSE [24]. Then, the bacterial and fungal OTUs were classified through BLAST in the Ribosomal Database Project (RDP) [25] and Unite [26] databases, respectively. The chimeric sequences were detected and removed using USEARCH 6.1 in QIIME [27]. To avoid bias due to sequencing depth, all samples were subsampled based on the minimum number of bacterial and fungal sequencing depth of this study. All raw sequences were deposited in the NCBI Sequence Read Archive with the submission Accession Number SRP 156,811 and SRP 156,814.

### 2.7. Statistical Analysis

The alpha diversity indices were calculated in QIIME. The data of soil physichemical properties, bacterial and fungal total abundances, alpha diversity indices, bacterial and fungal taxa (phyla and genera) in monoculture and intercropping systems were compared by Student’s t-test. All analyses were performed using SPSS software (Version 22.0, Armonk, NY, USA).

The response ratio (RR) was used to determine the changes of bacterial and fungal OTUs in the intercropping system compared to the monoculture system with a 95% confidence interval [28], and forest plots were created using SigmaPlot software (Version 12.5, Santa Clara county, CA, USA ). The linear discriminant analysis (LDA) effect size (LefSe) method ensured the selection of the top 40 abundant bacterial and fungal genera that were significantly associated with monoculture and intercropping systems. The alpha value employed for the factorial Kruskal–Wallis test was 0.05 (using all-against-all comparisons), and the threshold employed on the logarithmic LDA score for discriminative features was 2.0. Beta diversity and was performed based on taxonomic and phylogenetic measurements using Bray–Curtis distance. The relationship between the soil microbial communities and soil physicochemical properties was further analyzed using redundancy analysis (RDA) and the Mantel test. The beta diversity, RDA and the Mantel test mentioned above were performed using the “vegan” package in the R environment (R Version.3.2.0, Vienna, Austria) [29].

## 3. Results

### 3.1. Tomato Seedling Dry Biomasses and Soil Physicochemical Properties

The shoot and root dry biomasses of tomato seedlings and soil physicochemical properties of the monoculture and intercropping systems were summarized in Table 1. Tomato seedlings grown in the tomato/potato-onion intercropping system had significantly higher shoot and root dry biomasses than tomato monoculture system (*p* < 0.05). Compared with the monoculture system, soil pH and SOC were significantly increased and soil EC, AP and inorganic N contents were significantly decreased in the intercropping system (*p* < 0.05). No significant differences between the two systems were found in soil AK and moisture content (*p* > 0.05).

### 3.2. Soil Microbial Community Abundance

Abundance of soil bacterial community was significantly higher in the intercropping system than that in the monoculture system (*p* < 0.05) (Figure 1a), while the abundance of soil fungal community had no difference between the two systems (*p* > 0.05) (Figure 1b).

### 3.3. Amplicon Sequencing Data

A total of 222,849 quality 16S rRNA sequences and 223,171 quality ITS sequences were obtained from all soil samples from the monoculture and intercropping systems, and the sequences were grouped into 1281 bacterial OTUs and 667 fungal OTUs. The average length of the bacterial and fungal reads was 396 bp and 256 bp, respectively. The Good’s coverage, which reflects the captured diversity, was larger than 98.99% for all samples (data not shown).

### 3.4. Soil Microbial Community Diversities and Structures

The numbers of OTUs and Shannon and inverse Simpson indices of both the soil bacterial and fungal communities were significantly higher in the intercropping system than those in the monoculture system (*p* < 0.05) (Figure 2). These results show that the alpha diversities of the soil bacterial and fungal communities in all samples were significantly increased by intercropping with potato-onion.

The NMDS analysis showed that the stress values of bacterial and fungal communities were both 0.00 (Figure 3a,b), and the NMDS plots of bacterial community at the OTU level showed a clear distinction between the monoculture and intercropping systems (Figure 3a). The beta diversity based on Bray–Curtis demonstrated that soil bacterial community structure was significantly changed by the tomato/potato-onion intercropping system (*p* < 0.05) (Figure 3c), while the fungal community exhibited no statistically significant difference between the two cropping systems (*p* > 0.05) (Figure 3d).

### 3.5. Soil Bacterial Community Composition

Twenty-four bacterial phyla were found across all soil samples. Bacterial OTUs were predominantly associated with the phyla Proteobacteria, Actinobacteria, Cyanobacteria, Acidobacteria and Bacteroidetes (relative abundance > 5%), and these five phyla accounted for 90.98% and 85.57% (mean % across all treatment groups) of the total bacterial sequences in the monoculture and intercropping systems, respectively (Figure 4a). However, these were variations in relation abundances of these phyla between the two cropping systems. Compared with the monoculture system, the intercropping system had significantly higher relative abundances of Actinobacteria, Cyanobacteria, Acidobacteria, Firmicutes, Gemmatimonadetes, Chloroflexi, Planctomycetes, Deltaproteobacteria and Nitrospirae, and lower relative abundances of Alphaproteobacteria and Bacteroidetes (*p* < 0.05).

Taxonomical classification revealed that 396 bacterial genera were detected at the bacterial genus level in this study (Appendix A). Among the top 20 classified bacterial genera, the relative abundances of *Sphingobium*, *Flavobacterium*, *Lysobacter*, *Variovorax* and *Dyadobacter* were lower (*p* < 0.05), while the relative abundances of *Subgroup-6*, *Arthrobacter*, *Bacillus*, *Pseudomonas*, *Gemmatimonas*, *Devosia*, *Nitrospira*, *Bradyrhizobium* and *Bryobacter* were significantly higher in the intercropping system than those of the monoculture system (*p* < 0.05) (Figure 4c).

Histogram of the LAD scores showed that 15 genera of *Subgroup-6*, *Arthrobacter*, *Bacillus*, *Pseudomonas*, and *Nitrospira*, etc., were more abundant in the intercropping system, and *Sphingobium*, *Flavobacterium*, *Variovorax*, and *Dyadobacter* etc. were more abundant in the monoculture system. Among them, *Subgroup-6* was the most dominant genus in the intercropping system, whereas *Sphingobium* was hyper-dominant in the monoculture system, accounting for 24.3% of the total bacterial sequences in the monoculture system (only 3.71% in the intercropping system) (Figure 5a).

The relative abundances of the top 20 classified bacterial OTUs were selected to further illustrate the impact of intercropping on the composition of the soil bacterial community. It showed a 95% confidence interval compared with the monoculture system. According to the RR analysis, we found that the top 20 bacterial OTUs mainly belonged to the bacterial phyla Proteobacteria, Acidobacteria, Actinobacteria, Bacteroidetes and Firmicutes and observed either an increase or decrease in bacterial OTUs in monoculture and intercropping systems, as shown in Figure 6a.

### 3.6. Soil Fungal Community Composition

The dominant fungal phyla were Ascomycota, Basidiomycota and Zygomycota (relative abundance >5%) in the monoculture and intercropping systems, and these three phyla accounted for 95.1% and 90.6% of the total fungal sequences, respectively. Moreover, the relative abundances of the phylum Chytridiomycota were 0.32% and 2.85% in the monoculture and intercropping systems, respectively. A few other phyla were also found in all soil samples, including Fungi_unclassified and Glomeromycota, and the relative abundances were 4.61% and 6.56%, respectively (mean % across all treatment groups). Among all phyla, the relative abundances of Chytridiomycota were significantly higher in the intercropping system than those in the monoculture system (*p* < 0.05). The relative abundances of Ascomycota, Basidiomycota, Zygomycota and other phyla varied; however, there were no significant differences between the monoculture and intercropping systems (*p* > 0.05) (Figure 4b).

Taxonomical classification revealed that 219 fungal genera were detected at the fungal genus level in this study (Appendix A). In the top 20 fungal genera with the highest abundance, the intercropping system had higher relative abundances of *Olpidium* and *Cryptococcus* and lower relative abundance of *Cladosporium* compared with the monoculture system (*p* < 0.05) (Figure 4d).

The histogram of the LAD scores of the 40 most abundant fungal genera showed that *Olpidium*, *Torula*, *Stachybotrys* and Mortierellaoeae-unclassified were more abundant genera in the intercropping system, whereas Davidiellaceae-unclassified was a prominent genus in the monoculture system (Figure 5b).

The top 20 fungal OTUs were selected for the RR analysis at a 95% confidence interval, and they mainly belonged to Ascomycota, Basidiomycota, Chytridiomycota and Zygomycota and were observed to either increase or decrease with fungal OTUs in the monoculture and intercropping systems (Figure 6b).

### 3.7. Relationships between Soil Microbial Communities and Soil Physichemical Properties

Based on the results of the Mantel test, the soil bacterial community structure was strongly influenced by the soil pH (r = 0.707, *p* = 0.011) and SOC (r = 0.645, *p* = 0.044), but had no correlation with soil moisture (r = −0.289, *p* = 0.917), EC value (r = 0.682, *p* = 0.067), inorganic N (r = 0.082, *p* = 0.362), AP (r = 0.296, *p* = 0.165) and AK (r = −0.118, *p* = 0.687). For fungi, SOC (r = 0.493, *p* = 0.043) and AP (r = 0.489, *p* = 0.040) were significantly correlated with the fungal community structure, but not to the soil moisture (r = −0.216, *p* = 0.794), pH (r = 0.432, *p* = 0.06), EC value (r = 0.350, *p* = 0.093), inorganic N (r = 0.421, *p* = 0.057) and AK (r = 0.082, *p* = 0.455). The RDA showed the soil SOC and pH to be crucial environmental factors that were significantly correlated with the soil bacterial community structures of the samples (Figure 7a). The fungal community was strongly influenced by SOC and AP (Figure 7b).

## 4. Discussion

In this study, the shoot and root dry biomasses of tomato seedlings were significantly higher in the intercropping system than in the monoculture system at 45 d after tomato transplantation (Table 1), indicating our first hypothesis was fully validated. The result was also consistent with a previous study that found intercropping increased plant growth [15].

Plant diversity had profound effects on the soil environment [30]. Previous studies found that increased plant diversity could increase soil pH, organic carbon and nitrogen but decrease the soil EC value [17,31]. Similar results were observed in our study: The tomato/potato-onion intercropping system increased the soil pH and SOC, and decreased soil EC (Table 1). However, the soil inorganic N and AP contents were significantly lower in the intercropping system, which was inconsistent with another study [32]. It could be attributed to (1) the better tomato growth performance, which absorbed more nutrients, leading to less inorganic N and AP in the soil of the intercropping system and (2) there being more crops in the intercropping. Interestingly, no significant difference was observed for the soil AK in the two cropping systems (Table 1). This was likely due to the short-term period of the experiment. In general, these findings indicate that tomato with potato-onion intercropping system might improve soil environment by increasing soil carbon resource and lowering the levels of soil acidification and salinization, which could influence the soil microbial communities.

In this study, the RDA analysis (Figure 7) and Mantel test showed that the bacterial and fungal community structures were positively correlated with SOC. This suggests that soil carbon resources may be one of the key environmental factors in soil microbial community structure. This is consistent with the result observed by Lian et al. (2019) [33], who stated that the SOC in a sugarcane/soybean intercropping system play an important role in the change in soil microbial community structure. In addition, we found that other environmental factors had different effects on soil microbial communities. The fungal community structure was mainly driven by AP, which was consistent with the results from previous studies [34,35]. Moreover, the structure of the soil bacterial community was positively correlated with soil pH. This finding agrees with other studies that soil pH is one of the main factor that impacts bacterial community structure in agro-ecosystems because bacteria tend to be more sensitive than fungi to soil pH [36].

Generally, an increase in soil microbial diversity is beneficial to soil function and health [37]. In this study, the alpha diversity indices of bacterial and fungal communities were significantly higher in the intercropping system than those in the monoculture system (Figure 2). This result was consistent with previous findings that increasing plant diversity has a positive effect on soil microbial community alpha diversity [38]. The beta diversity and abundance of bacterial community were significantly higher in the intercropping system than in the monoculture system, while fungal community beta diversity and abundance exhibited no significant difference between the two cropping systems (Figure 3). These results indicated that our second hypothesis was not fully validated. Previous studies found that plant growth stages could be a factor in the soil fungal community [39]. For instance, similar fungal abundance was found at an early plant growth stage between mono- and inter-cropping systems, while significant difference was only observed at the maturing stage [40]. In short, intercropping with potato-onion increased soil microbial diversity, and these results might be another reason that higher tomato biomass was obtained in the tomato/potato-onion intercropping system.

In our study, we discovered that the soil microbial communities in the monoculture and intercropping systems exhibited unexpectedly high complexity. The bacterial communities across the two systems were dominated by the classes Alphaproteobacteria, Betaproteobacteria, Gammaproteobacteria and Deltaproteobacteria, and phyla Actinobacteria, Acidobacteria and Bacteroidetes (Figure 4a), which roughly correspond to the results of previous studies in agricultural ecosystems [41]. The RR analysis showed that the relative abundances of several bacterial OTUs increased or decreased with intercropping with potato-onion. Among the changed OTUs, many belonged to the class of Alphaproteobacteria and were categorized as *Sphingobium*, which was the dominant genus in the monoculture system (Figure 5 and Figure 6a)**.** Moreover, the relative abundances of *Lysobacter*, *Flavobacterium*, *Dyadobacter* and *Variovorax* were decreased in the intercropping system (Figure 4c). These taxa had the ability to utilize root metabolites, degrade aromatic compounds and decompose crop residues [42,43,44,45,46]. In summary, these results indicated that intercropping with potato-onion might reduce the ability to degrade lignin, lignin-derived aromatic compounds and/or root metabolites. Presumably, species-specific effects of plants on soil microbial communities through a variety of plant-derived materials, such as plant litter and root exudates, could generate strong microbial activities and interactions and contribute to specific taxa with degraded plant lignocellulose and/or organic compound potential [47]. These effects might be the reason for the differences in soil community compositions between monoculture and intercropping systems. Another reason could be explained by the relative abundances of soil microorganisms being affected by other biotic factors and various environmental factors [48].

LefSe analysis (Figure 5a) revealed that *Subgroup-6* was the dominant category of genus in the intercropping system with higher pH. This result was in line with previous studies that found that the relative abundance of *Subgroup-6* was positively correlated with pH [49]. Soil microbial taxa with the potential to involve nutrient cycling, inhibit soil-borne pathogens and/or promote plant growth such as *Arthrobacter*, *Bacillus*, *Pseudomonas*, *Nitrospira* and *Cryptococcus* [50] were enriched, while some soil pathogens such as Davidiellaceae-unclassified [51] and *Cladosporium* [52] were decreased in the tomato/potato-onion intercropping system. In addition, while the relative abundance of *Chaetomium* exhibited no difference between the monoculture and intercropping systems, the RR analysis (Figure 6b) showed that several OTUs that were categorized as *Chaetomium* consistently increased in the intercropping system. This finding suggested that *Chaetomium* might play a role in antagonizing the soil pathogens in an intercropping system [53]. These results were consistent with previous findings that diversified cropping systems could increase the relative abundance of some taxa with potential nutrient cycling, pathogen-antagonistic and/or plant growth promotion functions and decrease the relative abundance of some potential soil pathogen taxa [54], which supports our third hypothesis. Overall, a tomato/potato-onion intercropping system might be used in agriculture practice because intercropping with potato-onion could stimulate soil microbial communities that are beneficial to plant growth. Remarkably, tomato and potato-onion tend to be dependent on arbuscular mycorrhizal fungi (AMF), but AMF families were not amplified in high proportion in two cropping system. This may be due to poor amplification of AMF with general fungal ITS primers [55]. In this regard, more suitable primers should be selected in future studies, which could be better for investigating the diversity and composition of the soil microbial community in diversified cropping systems. In addition, while soil pathogens (such as *Fusarium* and *Cladosporium*) were detected in both cropping systems, none of the tomato plants showed signs of disease. This latter observation was possibly due to the antagonistic effects of other soil microorganisms or, more likely, to the short-term nature of the experiment. Therefore, long-term experiments are needed to evaluate the effects of potato-onion intercropping on soil-borne pathogens in the future.

## 5. Conclusions

Taken together, the results from this study indicate that the intercropping system increased tomato seedling biomass, alpha diversity of soil microbial community, abundance and beta diversity of bacterial community. The structure of the soil microbial community was significantly influenced by SOC, and bacterial and fungal communities were affected by pH and AP, respectively. Besides, the intercropping system had positive effects on soil bacterial and fungal communities by promoting beneficial bacteria and fungi and reducing certain potential plant pathogens. Further research will focus on the functions of special certain taxa and the relationship between soil physicochemical properties and special certain taxa in intercropping systems.

## Figures and Tables

**Figure 1 microorganisms-08-00834-f001:**
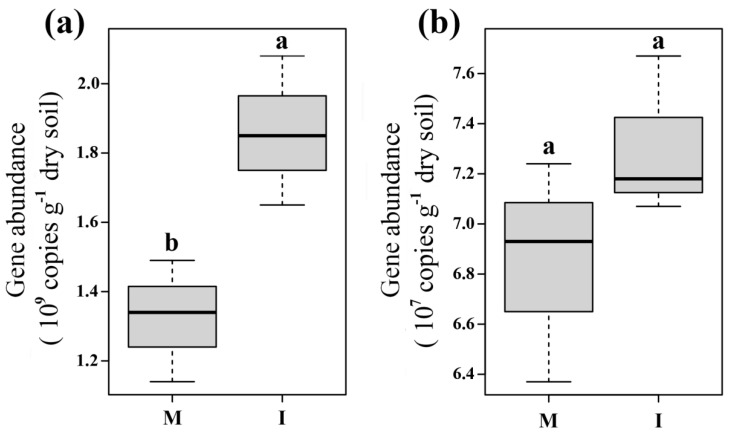
Soil microbial abundances of bacteria (**a**) and fungi (**b**) in monoculture (M) and intercropping (I) systems. Different letters indicate significant differences (*p* < 0.05; Student’s *t*-test.).

**Figure 2 microorganisms-08-00834-f002:**
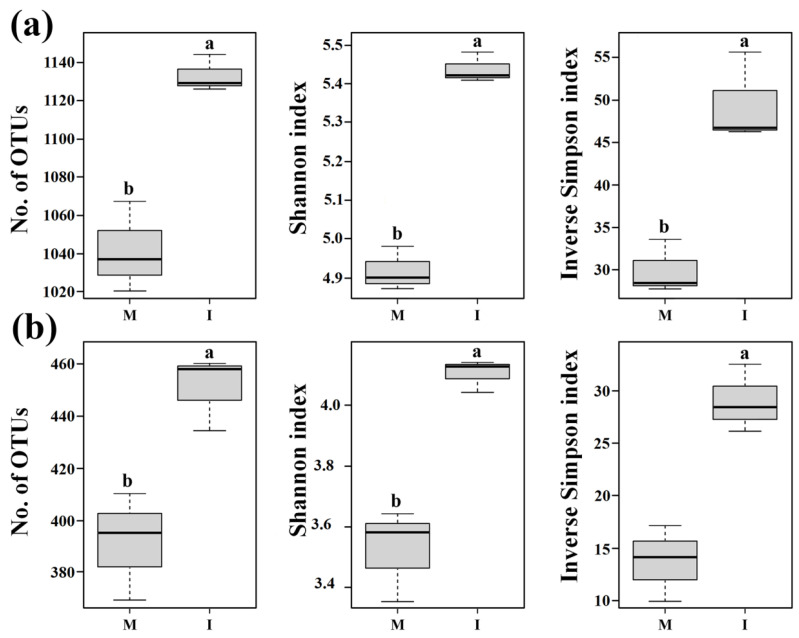
Alpha diversity based on Illumina MiSeq sequencing of bacterial (**a**) and fungal (**b**) communities in tomato monoculture (M) and intercropping (I) systems. Operational taxonomic units (OTUs) were delineated at 97% sequence similarity. Different letters indicate significant differences (*p* < 0.05; Student’s t-test.).

**Figure 3 microorganisms-08-00834-f003:**
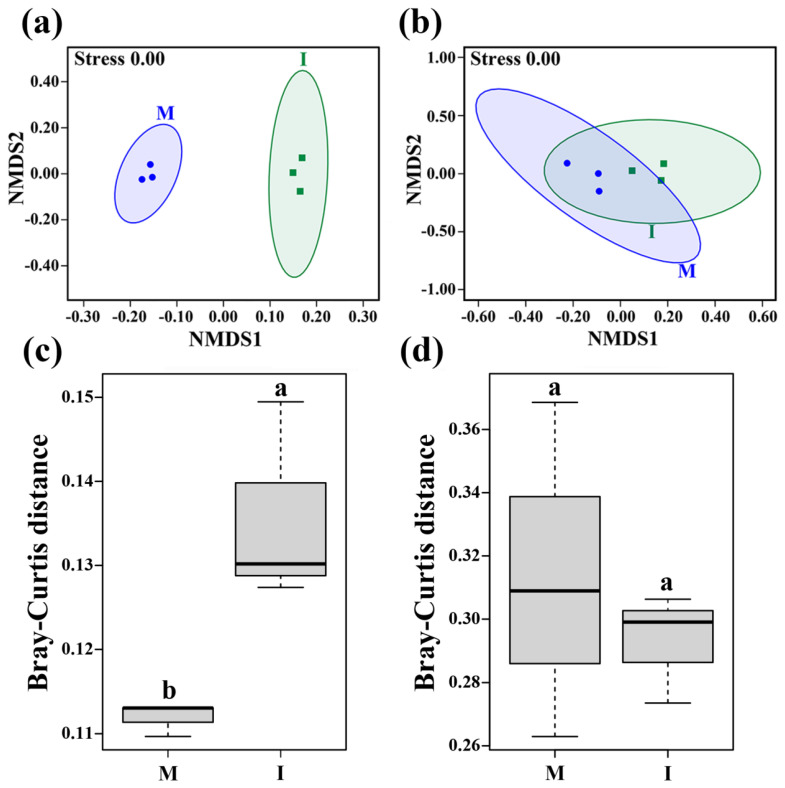
Beta diversity of bacterial (**a**,**c**) and fungal (**b**,**d**) communities in the monoculture (M) and intercropping systems (I) based on Bray–Curtis distances at the OTU level. Different letters indicate significant differences (*p* < 0.05; Student’s *t*-test.).

**Figure 4 microorganisms-08-00834-f004:**
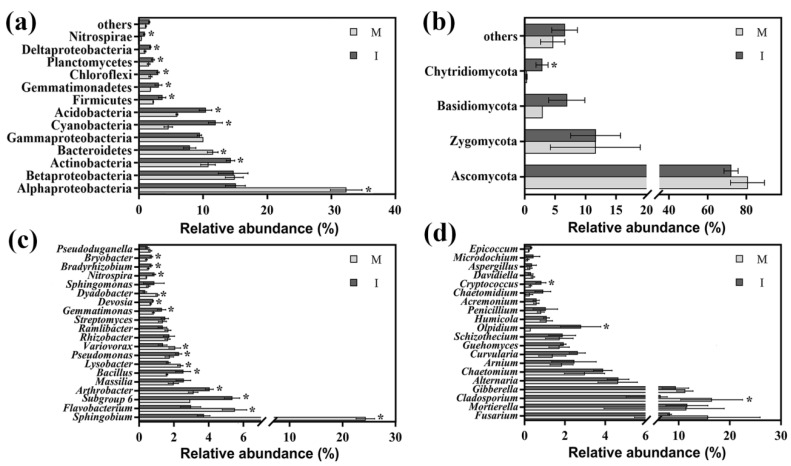
Relative abundances of the main bacterial phyla and proteobacterial classes (**a**) and fungal phyla (**b**) and the top 20 classified bacterial genera (**c**) and fungal genera (**d**) in monoculture (M) and intercropping (I) systems. * represents significance (*p* < 0.05) between the soil samples from the monoculture and intercropping systems according to Student’s *t*-test.

**Figure 5 microorganisms-08-00834-f005:**
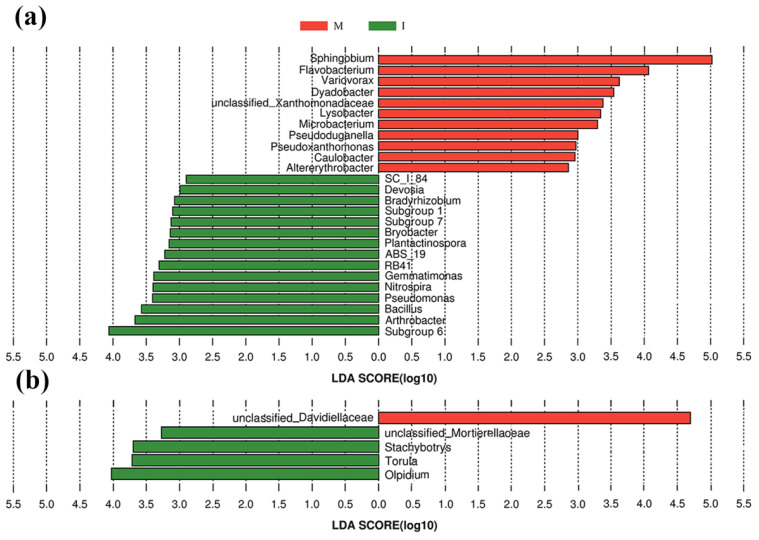
Histogram of the linear discriminant analysis (LDA) scores computed for differentially abundant bacterial (**a**) and fungal (**b**) genera between the monoculture system (M) and intercropping system (I). The threshold employed on the logarithmic LDA score for discriminative features was 2.0.

**Figure 6 microorganisms-08-00834-f006:**
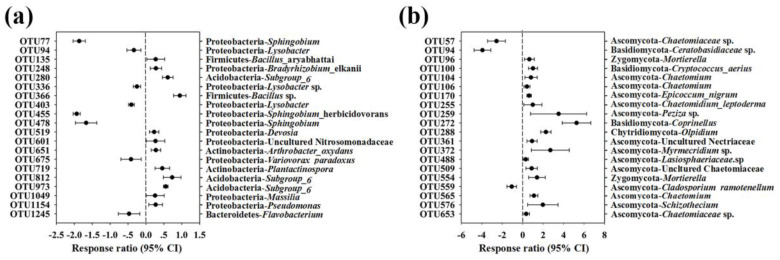
Response ratio analysis of changes in the relative abundances of the top 20 classified bacterial OTUs (**a**) and fungal OTUs (**b**) in response to intercropping with potato-onion compared to the monoculture system at 95% confidence interval. Error bars plotted to the right of the dashed line indicate that the relative abundance increased, while those on the left side indicate that the relative abundance decreased.

**Figure 7 microorganisms-08-00834-f007:**
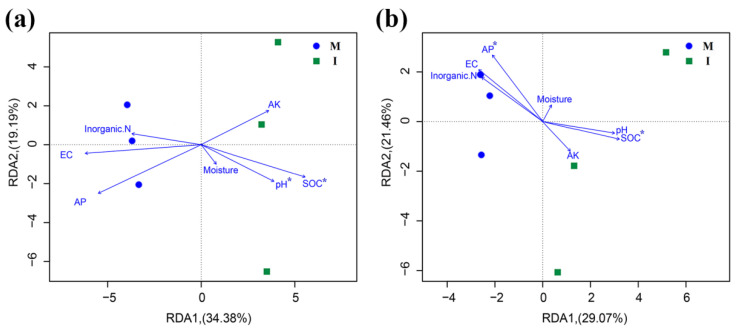
Redundancy analysis (RDA) depicting the relationship between the bacterial (**a**) and fungal (**b**) communities and environmental variables in monoculture (M) and intercropping (I) systems.

**Table 1 microorganisms-08-00834-t001:** Soil physicochemical properties in monoculture (M) and intercropping (I) systems.

Treatments	Tomato Shoot Dry Biomass (g Plant^−1^)	Tomato Root Dry Biomass (g Plant^−1^)	Moisture (%)	Soil pH	EC ^a^ (mS cm^−1^)	Inorganic N ^a^ (mg kg^−1^)	SOC ^a^ (g kg^−1^)	AP^a ^ (mg kg^−1^)	AK ^a^ (mg kg^−1^)
M	25.32 b	2.32 b	21.33 a	6.41 b	0.65 a	140.44 a	34.36 b	90.17 a	209.64 a
I	29.96 a	2.67 a	22.01 a	6.61 a	0.55 b	127.59 b	45.68 a	74.30 b	215.31 a

Values are mean ± standard deviation, and the valves followed by different letters for a given factor are significantly different (*p* < 0.05; Student’s *t*-test.). ^a^ EC, Inorganic N, SOC, AP and AK indicate soil electrical conductivity, inorganic nitrogen, soil organic carbon, available phosphorus and available potassium, respectively.

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
