# Peer review of "Intercropping with Potato-Onion Enhanced the Soil Microbial Diversity of Tomato"

_microorganisms, 2020, doi:10.3390/microorganisms8060834_

Round 1
Reviewer 1 Report
In this manuscript, Li and collaborators describe the effects of tomato/potato-onion intercropping systems on tomato seedlings growth and changes of soil microbial communities. The authors showed that the intercropping increased tomato seedling biomass, as well as the diversity alpha and beta community, that positively influenced plant growth. The analysis indicated that bacterial community structure was significantly influenced by soil organic carbon and pH, the fungal community structure was related to changes in soil organic carbon and available phosphorus.
Overall, the study provides interesting results regarding the positive effect of bacterial and fungi soil on plant growth. Interestingly, a large set of data was collected that provides a wide picture of the plant status under these conditions.
The paper, however, must be improved in terms of writing since some grammar and syntax errors are present in the manuscript. Also, a better description of the results should be considered. Furthermore, the discussion section must be more focused on your own data. For instance, explain how all the variables that were measured correlate among each other in light of the observed results? only due to the availability of organic carbon, changes in pH and phosphorus availability? How do you explain that the positive effect of the microbes is only observed under growth? etc. I kindly suggest reorganizing and shortening this section.
Minor comments:
In the introduction it would be interesting to mention the direct and indirect positive effects of microorganisms that promote plant growth.
In figure 4a, I suggest the breakdown of the set of proteobacteria, in: alpha, beta, gamma, delta and epsilon.
Lines 235-236: I suggest presenting in supplementary material (bacterial genera).
Lines 281-282: I suggest presenting in supplementary material (fungal genera).
Line 325: I suggest transferring table 1 to the results section.
Line 353: What kind of Proteobacteria?
Conclusions: I suggest removing the phrase: Tomato / potato-onion intercropping system increased the relative abundances of some bacterial and fungal taxa (i.e., Subgroup-6 Arthrobacter, Bacillus, Pseudomonas and Cryptococcus) which possess pathogen-antagonistic and / or plant growth promoting potentials. Moreover, the relative abundance of some taxa (i.e., Cladosporium and Davidiellaceae) containing potential pathogens decreased. This account must be in the results or in the discussion.
Author Response
Open Review
Comments and Suggestions for Authors
In this manuscript, Li and collaborators describe the effects of tomato/potato-onion intercropping systems on tomato seedlings growth and changes of soil microbial communities. The authors showed that the intercropping increased tomato seedling biomass, as well as the diversity alpha and beta community, that positively influenced plant growth. The analysis indicated that bacterial community structure was significantly influenced by soil organic carbon and pH, the fungal community structure was related to changes in soil organic carbon and available phosphorus.
Overall, the study provides interesting results regarding the positive effect of bacterial and fungi soil on plant growth. Interestingly, a large set of data was collected that provides a wide picture of the plant status under these conditions.
The paper, however, must be improved in terms of writing since some grammar and syntax errors are present in the manuscript. Also, a better description of the results should be considered. Furthermore, the discussion section must be more focused on your own data. For instance, explain how all the variables that were measured correlate among each other in light of the observed results? only due to the availability of organic carbon, changes in pH and phosphorus availability? How do you explain that the positive effect of the microbes is only observed under growth? etc. I kindly suggest reorganizing and shortening this section.
Response: We highly appreciate your constructive comments to help us improve this manuscript. We have carefully modified the manuscript according to your specific comments. The language of this manuscript have been polished and the results portion have been rewritten. In addition, the discussion was reorganized to focus more on the data in this study, including the linkage between tomato growth and community, the soil microbial communities and soil physichemical properties.
Minor comments:
In the introduction it would be interesting to mention the direct and indirect positive effects of microorganisms that promote plant growth.
Response: Thanks very much for the valuable comments and significant guidance. As suggested, several reports on some soil microbial taxa directly or indirectly affect soil productivity by producing plant growth hormones, solubilizing soil P and enhancing nitrogen fixation are now cited [8-13] and supplemented in the introduction. Related descriptions are now added in the revised manuscript. (Page 1-2, line 40-45;).
In figure 4a, I suggest the breakdown of the set of proteobacteria, in: alpha, beta, gamma, delta and epsilon.
Response: Thanks very much for the valuable comment and kind suggestion,in new figure 4a the breakdown of the set of proteobacteria, in: alpha, beta, gamma, delta. Related descriptions were revised in the manuscript. (Page 8, Fig 4)
Lines 235-236: I suggest presenting in supplementary material (bacterial genera).
Response: Thanks very much for the considerate review and kind suggestion. The relative abundances of the bacterial genera in in monoculture (M) and intercropping (I) systems (average relative abundances >0.30% in at least one treatment) have been added in the supplementary material. (Page 18, added Table S1) .
Lines 281-282: I suggest presenting in supplementary material (fungal genera).
Response: Thanks very much for the considerate review and kind suggestion. The relative abundances of the fungal genera in in monoculture (M) and intercropping (I) systems (average relative abundances >0.30% in at least one treatment) have been added in the supplementary material. (Page 19, added Table S2).
Line 325: I suggest transferring table 1 to the results section.
Response: Thanks very much for the valuable comment and kind suggestion. The table 1 has been transferred to the results section (Page 5, line 187-188).
Line 353: What kind of Proteobacteria?
Response: Thanks very much for the considerate review and kind suggestion. In our study, the bacterial communities across the two systems were dominated by the classes alphaproteobacteria, betaproteobacteria, gammaproteobacteria, deltaproteobacteria, and phyla Actinobacteria, Acidobacteria, and Bacteroidetes in agricultural ecosystems. Related descriptions were revised in the manuscript. (Page 11, line 351-354)
Conclusions: I suggest removing the phrase: Tomato / potato-onion intercropping system increased the relative abundances of some bacterial and fungal taxa (i.e., Subgroup-6 Arthrobacter, Bacillus, Pseudomonas and Cryptococcus) which possess pathogen-antagonistic and / or plant growth promoting potentials. Moreover, the relative abundance of some taxa (i.e., Cladosporium and Davidiellaceae) containing potential pathogens decreased. This account must be in the results or in the discussion.
Response: Thanks very much for the considerate review and kind suggestion. We have removed this phrase and rewritten the conclusions (Page 12, line 398-404).
Reviewer 2 Report
The MS report the effect on tomato seedlings of intercropping them with potato-onion. Tomato plants cultivated in monoculture and intercropped were compared in some growth parameters. In addition, by mean of NGS the soil microbial community (Fungal and bacterial ones) associated with the 2 different managements was studied and compared.
This specific argument, although not really new, is often contradictory and fragmented among studies and sometimes not so deeply studied, thus this MS could be of interest for people working on horticulture, both in research and applied aspects.
However, I have serious doubts about the research.
It seems to me that the most severe problem is not having included and analyzed by NGS the soil before transplanting tomatoes (in monoculture or intercropping) thus not have idea of the starting microbial communities present at time 0 to be compared with both the treatments.
The English is very poor along the entire MS and there are several type-errors and words used inappropriately (e.g “diversities” in the title, Illuminate instead of Illumina MiSeq ).
Some important information about the experiment are lacking:
For example: the length of the experiment is not clear and it has reported only in the results and not in M&M. (45 days???)
Another example the construction of the standard curve for RTPCR...what kind of standard (plasmid carrying the target DNA....) have been used?
Why the authors have not considered to separate roots and aerial part when measuring the DW? I think this information might have a certain importance in relation to the retrieved soil microbial communities and their possible effects.
I am not surprised that AM fungi were not amplified in high proportion because the couple of primers used have strong bias toward them. However this point need to be discuss and cite in the results and discussion because both tomatoes and onion are very well mycorrhized plants and these fungi could represent a very important fraction of horticultural soil fungal communities exerting important influence as biofertilizer.
This is very important and deserve to be investigated because is surely a key factor to be considered when not only bacteria but also soil fungal population is under study.
The literature review in the introduction and discussion must be improved and I think it is quite short considering the topic of the study.
Author Response
Open Review
Comments and Suggestions for Authors
The MS report the effect on tomato seedlings of intercropping them with potato-onion. Tomato plants cultivated in monoculture and intercropped were compared in some growth parameters. In addition, by mean of NGS the soil microbial community (Fungal and bacterial ones) associated with the 2 different managements was studied and compared.
This specific argument, although not really new, is often contradictory and fragmented among studies and sometimes not so deeply studied, thus this MS could be of interest for people working on horticulture, both in research and applied aspects.
However, I have serious doubts about the research.
It seems to me that the most severe problem is not having included and analyzed by NGS the soil before transplanting tomatoes (in monoculture or intercropping) thus not have idea of the starting microbial communities present at time 0 to be compared with both the treatments.
Response: We highly appreciate your constructive comments to help us improve this manuscript. We have carefully modified the manuscript according to your specific comments. The aim of this study is to investigate the change of soil microbial community in the tomato monoculture system and tomato/potato-onion intercropping system. Therefore, we did not measure the changes of soil microbial community before transplantation. However, in the future studies, we will focus on the changes of soil microorganisms before and after planting under monoculture and intercropping systems.
The English is very poor along the entire MS and there are several type-errors and words used inappropriately (e.g “diversities” in the title, Illuminate instead of Illumina MiSeq ).
Response: Thanks very much for the kind correction. We apologize for our poor writing and the language of the manuscript have been polished. Such as title has been revised to “Intercropping with Potato-Onion Enhanced the Soil Microbial Diversity of Tomato”.
Some important information about the experiment are lacking:
For example: the length of the experiment is not clear and it has reported only in the results and not in M&M. (45 days???)
Response: Thanks very much for the valuable comments and significant guidance. The length of the experiment was added in M&M (Page 2, line 87-88)
Another example the construction of the standard curve for RTPCR...what kind of standard (plasmid carrying the target DNA....) have been used?
Response: Thanks very much for the valuable comments and significant guidance. In this study, the standard curves of bacterial and fungal were created with 10-fold dilution series of plasmids containing the target genes from soil samples. Related descriptions have been revised in the manuscript. (Page 3, line 129-130)
Why the authors have not considered to separate roots and aerial part when measuring the DW? I think this information might have a certain importance in relation to the retrieved soil microbial communities and their possible effects.
Response: Thanks very much for the valuable comments and significant guidance. In the revised manuscript, the shoot and root dry biomasses of the tomato seedlinghave been separated, the data were showed in Table 1, and we have modified the corresponding results and discussion portion.
I am not surprised that AM fungi were not amplified in high proportion because the couple of primers used have strong bias toward them. However this point need to be discuss and cite in the results and discussion because both tomatoes and onion are very well mycorrhized plants and these fungi could represent a very important fraction of horticultural soil fungal communities exerting important influence as biofertilizer.
This is very important and deserve to be investigated because is surely a key factor to be considered when not only bacteria but also soil fungal population is under study.
Response: According to your suggestion, the phenomenon that AMF were not amplified in high proportion in the two cropping systems have been further discussed in the discussion portion as follows: “Remarkably, Tomato and potato-onion tend to be dependent on arbuscular mycorrhizal fungi (AMF), but AMF families were not amplified in high proportion in two cropping system. This may be due to poor amplification of AMF with general fungal ITS primers [58]. In this regard, more suitable primers should be selected in future studies, which could be better to investigate the diversity and composition of soil microbial community in diversified cropping systems.” (Page 12, line 386-391).
The literature review in the introduction and discussion must be improved and I think it is quite short considering the topic of the study.
Response: We have improved the introduction and discussion according to your comments to make them more clear and consider the topic.
Round 2
Reviewer 2 Report
The authors have tried to meet and respond to my comments
on the first version of the MS.
I'm still not fully satisfied but the MS in this version has certainly been improved.
There are still some inaccurancies to be corrected:
Such as: line 13-15 “Results showed that the intercropping system increased shoot and root biomasses of tomato seedlings,as well as the alpha-diversity of soil microbial communities and abundance and beta-diversity of sole bacterial community. Again type errors such as Illuminate instead of Illumina both in: keywords and line 59 of Introduction. Line 42 please correct taxa with taxa/group , because just below you describe bacterial taxa but also a type of fungal group (the AMF). Line 44-45 Please use: arbuscular mycorrhizal fungi (AMF) or Arbuscular Mycorrhizal Fungi (AMF). 325 I suppose Mantel not Mental
Author Response
Dear Reviewer,
We highly appreciate your constructive comments to help us improve this manuscript. We have carefully modified the manuscript according to your specific comments.
#Line 13-15 “Results showed that the intercropping system increased shoot and root biomasses of tomato seedlings, as well as the alpha-diversity of soil microbial communities and abundance and beta-diversity of sole bacterial community.”
Response: Thanks very much for the valuable comments and significant guidance. In the revised manuscript, the related descriptions have been revised in the manuscript in line 13-18 as follows:“Results showed that the intercropping with potato-onion increased tomato seedling biomass. Compared with monoculture system, the alpha diversity of soil bacterial and fungal communities, beta diversity and abundance of bacterial community were increased in the intercropping system. Nevertheless, the beta-diversity and abundance of fungal community had no difference between the intercropping and monoculture systems”.
#Line 42 please correct taxa with taxa/group , because just below you describe bacterial taxa but also a type of fungal group (the AMF).
Response: Thanks very much for the valuable comments and significant guidance. In the revised manuscript, the related descriptions have been revised in the manuscript in line 45 as follows :“ some taxa”
#Line 44-45 Please use: arbuscular mycorrhizal fungi (AMF) or Arbuscular Mycorrhizal Fungi (AMF). 325 I suppose Mantel not Mental
Response: Thanks very much for the valuable comments and significant guidance. In the revised manuscript, the related descriptions have been revised in the manuscript in line 47, and line 327.
#Again type errors such as Illuminate instead of Illumina both in: keywords and line 59 of Introduction.
Response: Thanks very much for your valuable comments. After careful consideration, we recommend the pristine “Illumina MiSeq Sequencing” here, which were same as the highly cited papers, as follows:
Wei H , Wang L , Hassan M , et al. Succession of the functional microbial communities and the metabolic functions in maize straw composting process[J]. Bioresource Technology, 2018:S0960852418302347.
R, Henrik, Nilsson , et al. Mycobiome diversity: high-throughput sequencing and identification of fungi.[J]. Nature Reviews Microbiology, 2018, 17, 95–109.
Yao Q , Liu J , Yu Z , et al. Three years of biochar amendment alters soil physiochemical properties and fungal community composition in a black soil of northeast China[J]. Soil Biology and Biochemistry, 2017, 110:56-67.
James J. Kozich , Sarah L. Westcott , et al. Applied and Environmental Microbiology Aug 2013, 79 (17) 5112-5120.
We also carefully checked and modified the language, spelling, and grammar throughout the manuscript as follows:
For Introduction, we changed lines 33-34, as follow: Agricultural studies have found that intercropping can enhance biodiversity by increasing spatial and temporal plant diversity” and line 36: added “characterized by monoculture”.
For Materials and methods, we changed Line 88 “Plant and soil sampling” instead of “Plant and Soil sampling” and line 162, “SRP 156,811 and SRP 156,814” instead of “SRP 15811 and SRP 156814”.
For Results, we changed line 205 “Shannon and inverse Simpson indices” instead of “Shannon index and inverse Simpson index”, line 243 “than those of monoculture system” instead of “than that of monoculture system”.
For Discussion, we added “soil carbon resources” at line 328 and “involve nutrient cycling” at line 375, changed line 339 “alpha diversity indices” instead of “alpha diversities indices”
For Conclusions, we changed lines 402-403, as follow: “bacterial and fungal communities were affected by pH and AP, respectively”.